# Tackling Food Waste: Impact of German Consumer Behaviour on Food in Chilled Storage

**DOI:** 10.3390/foods9101462

**Published:** 2020-10-14

**Authors:** Harald Wucher, Astrid Klingshirn, Lilla Brugger, Rainer Stamminger, Jasmin Geppert, Beate Kölzer, Antje Engstler, Jochen Härlen

**Affiliations:** 1Department Life Sciences, University of Applied Sciences Albstadt-Sigmaringen, Anton-Guenther-Straße 51, 72488 Sigmaringen, Germany; klingshirn@hs-albsig.de (A.K.); brugger@hs-albsig.de (L.B.); 2Institute of Agricultural Engineering, Section Household and Appliance Technology, Bonn University, Nussallee 5, 53115 Bonn, Germany; stamminger@uni-bonn.de (R.S.); jasmin-geppert@t-online.de (J.G.); bkoelzer@uni-bonn.de (B.K.); 3BSH Hausgeraete GmbH, Robert-Bosch-Straße 100, 89537 Giengen an der Brenz, Germany; Antje.Engstler@bshg.com (A.E.); Jochen.Haerlen@bshg.com (J.H.)

**Keywords:** domestic refrigeration, consumer behaviour, food waste, food quality, food safety, storage features

## Abstract

Since perishable food in private households is usually stored in refrigerators, both consumer knowledge of correct storage parameters and food handling have a major impact on food quality retention. Today’s refrigerators provide features, such as climate-controlled compartments, to improve chilled storage performance, but former studies have shown a lack of usage and interest in such features. This paper is based on the results of an online survey with 2666 respondents from Germany. The questionnaire focused on consumer behaviour regarding the chilled storage of perishable foods, the perception of quality loss in food and consumer requirements concerning refrigerators. The novelty in the study layout was the focus on seven common refrigerator storage features. The results showed that consumers had a high awareness of food safety, resulting in short storage durations. While it was clearly indicated that the storage features were less well-known, their importance correlated with rates of food waste, underlining the potential for improvement. The survey implied that raising the awareness of storage climate parameters is a tool for nudging consumers into lower rates of food waste.

## 1. Introduction

The reduction of food losses at the end of the manufacturing chain, often defined as food waste, is a major topic in further evolving sustainable food management [1]. Private households, which contribute most to the avoidable waste in industrialized countries [2], are especially in focus in overall reduction strategies. Food waste in private households is a highly complex interaction of consumers’ behaviour, foods characteristics and technological development. The reasons for food waste from the types of food that show the highest proportions of waste and overall consumer attitudes towards food value and food handling itself have been analysed in detail throughout the last few years [2,3,4].

In 2017, average household-related food waste per capita in Europe was 79 kg [5]. The analysis of the influence of demographics show that food reduction behaviour is more likely for older age groups than for younger [6] and for female more likely than for male [7]. Tendencies vary between European countries based on the size of the household. Lower amounts of food waste are found in German, Finnish or Norwegian 1- or 2-person households than in larger households [3,6,8] while tendencies are reversed in the UK [9]. Consumer-related reasons for losses can mainly be traced back to cooking or preparing too much, wrong planning in terms of not using food in due time, not eating food before it exceeds its use-by date and improper handling of food [10]. Proper storage conditions and food processing in households have been found to be the central elements in food quality retention and sustainable food management for food-related reasons [11,12]. Quality retention in storage (Figure 1) is defined by food-specific parameters, such as the original food quality, time-temperature history, provision of accurate storage conditions [13] and the consumer competence in food handling and refrigerator performance [14].

The large gap in home economics skills, especially related to food management and refrigerator storage, is one of the root causes of avoidable food waste and safety problems [15,16]. Poor hygiene practices coupled with an overestimation of one’s own capabilities, limited sensitivity towards food safety issues in food storage and placement, and a lack of knowledge or negligence related to food-specific storage climate requirements and the correct refrigerator setting have been identified as major impairments to the freshness of food [17,18,19,20]. The knowledge of European consumers concerning the correct refrigerator storage temperature varies, with 71.6% in Ireland [21], 44.4% in Greece [22] and 13% in the UK [23]. It was shown for Germany that <50% of consumers adjust their refrigerators at recommended temperature settings [24]. Highly perishable foods requiring chilled storage are especially critical. Products that show the highest share in food waste are fresh fruit and vegetables, amounting to 30–50%, as reported by different European studies, followed by prepared foods, amounting to 15–30%. Fresh meat and fish make up 4%, however, and may not be neglected in importance due to the high primary resource input and their contribution to foodborne diseases [2,10,25]. Nearly 50% of food waste is rated as avoidable by consumers themselves, as shown by a recent German study [10], which underlines the potential of improved food management. At the same time, the real intention to change behaviour is limited: About 13% of consumers seem to be receptive towards recommendations and are willing to make some effort to reduce food waste. Nearly 60% are not receptive to the topic or are unconcerned [20]. Even though joint activities in food management optimisation in recent years, based on marketing campaigns such as “Love Food, Hate Waste” or “Think. Eat. Save”, have led to a first reduction in food waste per capita–95 kg a^−1^ reported in 2011 [2] vs. 79 kg a^−1^ reported in 2017 [5], there is still a huge potential for further reduction.

Consequently, two major input parameters should be considered and linked: the status quo in refrigerated food storage (simultaneously covering consumer food handling and refrigerator storage performance) and consumer expectations regarding food quality retention in storage. Studies analysing the consumer behaviour related to refrigerator storage that can be used as a first guidance provide information on storage temperature, storage places, storage time and packaging in storage. Measured refrigerator temperatures from different studies showed that the mean storage temperature in southern European countries is 7.0 °C (SD 2.7 K) and 6.1 °C (SD 2.8 K) in northern European countries [26]. Maximum temperatures varied between 10.0 and 21.3 °C, being much higher than the recommended storage temperature of 4 °C, especially regarding food safety aspects [17,26]. The mean temperature at the refrigerator’s middle shelves in Germany is 6.2 °C, within a range between 2.5 and 10 °C [27]. Regarding storage systematics, a Europe-wide study showed that distinct refrigerator storage places, such as vegetable drawers for the storage of fruit and vegetables, and the upper shelves and refrigerator door for dairy products, are well-adopted by end consumers. Meat, fish and leftovers are spread throughout the refrigerator shelves, with more than 50% of consumers following their own systematics, such as sorting food of the same category into one area [4]. A prediction of storage time identified the common rule that the storage time is described by an exponential distribution of the use-by date period divided by a common value equal to 4, which fits well for products with short use-by days, such as minced meat, fresh fish or ready-to-eat salads, which are stored by 75% of consumers for 1.4, 1.8 and 1.9 days, respectively [25,28]. Reported times for foods with a longer shelf life such as ham, salmon or cheese range from 6.3, 5.5 to 11.1 days, respectively.

Nowadays, storage performance in refrigerators is enhanced by various additional features that either augment storage climate parameters, such as temperature, humidity or gas atmosphere, or storage hygiene, such as odour control, antimicrobial air cleaning or surfaces [29,30]. Basic temperature definitions in chilled and frozen storage zones and more specific performance definitions for chill compartments provide a storage zone with temperatures from −2 to +3 °C. However, apart from this range, no further specifications related to freshness retention and storage performance are currently defined within International Electrotechnical Commission standard IEC 62552-2:2015, the standard specifying the essential characteristics of household refrigerating appliances [31]. Freshness and hygiene features whose performance parameters are not further specified and regulated include functions such as the inhibition of microbial growth by antimicrobial surfaces, air filtrations systems to reduce airborne microorganisms and odours or the exposure of fresh produce to blue light during storage to enhance the production of secondary plant metabolites [29]. Freshness features focus frequently on additional humidity control options, addressing transpiration losses in fruit and vegetables, which are especially enhanced in dynamic cooled appliances. The reduction of weight loss is crucial to prevent rapid product deterioration, with critical fresh weight loss starting from 3% in leafy vegetables [32]. Studies in the UK from 2016 showing that only 25% of the respondents are interested in technologies improving food freshness retention also showed that next to temperature, the importance of humidity control is also a parameter not well-known, as many consumers tend to intentionally open or leave packages open, assuming shelf life benefits [33,34]. Whether humidity control options within refrigerators are known, correctly used or recognized as valuable has not yet been analysed. At the same time, consumers’ quality expectations may not be neglected in order to be able to derive sound concepts in consumer guidance and storage optimisation. Food choice itself and the evaluation of suitability for consumption are made by visible and invisible cues, mostly directly or indirectly related to food quality and safety. Consumers are often not actively aware of the different cues, as the decision-making process is rather intuitive. Thus, forcing interaction and active consumer guidance and raising awareness is difficult. Major consumer-relevant food quality dimensions that must be maintained in storage are related to sensory quality, such as appearance, taste, smell and texture, and healthiness, which is defined mainly by safety, nutritive value and convenience [35]. A cross-European study showed that consumers find it hard to define these parameters strictly and, thus, often mix related attributes, especially in terms of quality and safety [36].

The main objective of this study is to assess which parameters may further promote an improvement of food management in refrigerator storage by passive strategies, thus, further optimising the consumer’s competence in food handling. We focus on a twofold approach: the encouraging of correct handling of the food categories fruit, vegetables, meat and fish on the part of the consumer, and the further improvement of the storage and feature layout of refrigerators on the part of refrigerator manufacturers. Therefore, the approach does not try to address food waste reduction directly, which imposes negativity and insinuates improper food handling and a lack of competence, but tries to strengthen already existing storage strategies and make use of food quality desires.

## 2. Materials and Methods

### 2.1. Consumer Survey

#### 2.1.1. Sampling Plan

An online-based consumer survey was conducted from 16 to 23 January 2018. The aim was to identify German consumers’ storage behaviour related to the highly perishable and chilled stored food categories fruit, vegetables, meat and fish, with a focus on the storage place, storage time, food waste and main reasons for waste, in order to derive relevant strategies. At the same time, the survey intended to uncover relevant storage quality parameters by the type of food, knowledge of existing refrigerator storage features and the relevancy rating of features by consumers to identify starting points for refrigerator layout optimisation. The respondents were recruited by a market research agency according to specific quota for gender, relevant age groups for the domestic refrigerator market (18–69) and household size, representing the average German population, yet not considering the quota for residential area [37,38,39]. The question of an urban or rural residential area was not taken into account in the demographic composition. The survey included one control question to verify the accuracy of the answers, resulting in 2666 valid respondents. In addition to the recruitment, the market research agency was responsible for programming and hosting the survey and quality assurance. Prerequisites for participation were the presence of at least one refrigerator in the household and full or joint responsibility for the food management of the respondent.

#### 2.1.2. Questionnaire

The questionnaire was divided into four sections: (1) demographics and appliance characteristics, (2) storage behaviour, (3) food waste and impact factors, and (4) consumer demands in refrigerator storage. Storage behaviour and food waste focused on the types of food: fruit, vegetables, meat and fish. The survey included 32 questions, some of which were underlined by pictograms and pictures for better understanding. Answer categories were predefined; the set-up for most questions was multiple-choice. If a respondent did not store a type of food, questions based on this type of food were excluded. A control question was included asking for the same information in two different ways to assure reasonable replies. The questionnaire took approximately 20 min to complete. Regarding the results concerning consumer behaviour, all possible answers per question are given along with the results and figures related. The questionnaire is shown in Appendix B.

### 2.2. Data Analysis

The statistical data analysis used the non-parametric test of the Spearman’s rank correlation coefficient (r_sp_), linear regression analyses and the two-way analysis of variance (ANOVA) for the test for between-subject effects. Spearmen’s rank correlation was chosen due to the ordinal level of measurement. The correlation was conducted to correlate the variables “food waste” and the food waste-related “loss of freshness” with variables that were related to domestic characteristics, consumers’ behaviour or demands. Linear regression analyses were conducted for the same variables to countercheck the results from the Spearman correlations, including ANOVA. The two-way ANOVA for the test for between-subject effects was conducted to verify the interactions between the variables’ possible impact on the dependent variable food waste and include no post hoc tests. The significance level for the test was 0.05. The questionnaire responses were analysed using Statistical Product and Service Solutions (SPSS) software, version 25.0 (IBM, Armonk, NY, USA) and Microsoft Excel 2016.

## 3. Results

### 3.1. Demographics and Appliance Characteristics

The demographic characteristics of the survey respondents and the key data on their refrigerator appliances are listed in Table 1. Demographic characteristics include gender, age of respondents as age groups, household size and the responsibility for the food management. Data on the refrigerator include the age of the refrigerator and the number of refrigerators in the household.

Gender distribution was almost 1:1. Almost 50% of the respondents lived in single-person households, approximately one third (30%) in two-person households and 22% in larger households. A total of 76% were fully responsible for the food management. Nearly half of the respondents used a second appliance in addition to the main refrigerator. More than 50% of the refrigerator models were younger than 5 years and 20% were older than 10 years.

### 3.2. Basic Consumer Behaviour in Refrigerator Storage

The food categories focused on were fruit, vegetables, meat and fish. The analysis of the refrigerator usage included the following requests or questions: “Indicate the average loading degree of your refrigerator compartment/vegetable drawer”, “How long do you usually store the different food categories until complete consumption?” and “Where in the refrigerator do you store the different food categories most often?”.

Answers for the loading degree of refrigerator compartments and vegetable drawers were “maximum”, “average”, “minimum” and “no load”. Regarding the compartment, 26.6% of the consumers reported a maximum load, 57.7% average, 15.3% minimum and 0.5% no load. Concerning the drawer, 23.6% of the consumers reported a maximum load, 47.7% average, 23.4% minimum and 5.3% no load. Consumer behaviour concerning storage time and place is listed in Table 2. The answers “14 days” and “more than 14 days” for storage time were summed up. The results for the storage place “door rack” are not shown due to percentages ≤1.

Fruit and vegetables were consumed within three to a maximum of five days by more than two-thirds of consumers. Extended storage times of more than one week were found in less than 8% of respondents. Meat products were usually consumed within three days (60%); a storage time of five to seven days was found in 34% of households. Fish products were consumed within a day of purchase by 37% of consumers; another 42% stored fish for a maximum of three days. Storage places showed distinct patterns per food category: fruit (43%) and especially vegetables (81%) were commonly stored in the vegetable drawer. A rather large percentage of fruit was stored in ambient conditions (30%). Meat and fish were commonly stored on the bottom or mid shelves.

### 3.3. Food Waste and Impact Factors

The analysis of storage problems focused on food waste and loss of freshness. Only respondents who claimed to store the specific food category were admitted regarding the question, “How often do you throw away the different food categories because they are no longer edible?” Half (50%) of the respondents never reported food waste for meat and about two-thirds (63%) for fish, while only about a quarter (22–25%) never faced food waste of fruit and vegetables. Around 6% reported weekly food waste within all food categories (Figure 2). 

Significant correlations for food waste are found for all food categories related to the consumer’s age, with lower food waste for older respondents, and for household size, with higher food waste for larger households, except for households with more than 5 people, which show the reverse tendency. Respondents who stored vegetables, meat and fish for longer durations reported lower rates of food waste (Table 3).

The ANOVA test from the linear regression analyses confirm the tendencies of the Spearman correlation with *p* = 0.000 for all tests except for the rate of food waste and the storage time of fruit with *p* = 0.784.

Only respondents who claimed to store the specific food category were admitted regarding the question, “In which foods do you notice loss of freshness?” The loss of freshness was defined by examples such as “vegetables wither” or “milk tastes sour”. A loss of freshness in fruit and vegetables was never observed was reported by 50–53% of the respondents and 40–42% reported a rare observance of a loss of freshness. A loss of freshness of meat and fish never occurring was reported by 62–64% of the respondents, and 31–33% stated that there was a rare loss of freshness. Less than 8% of the respondents indicated that loss of freshness was often experienced for all categories (Figure 3).

Significant correlations between the loss of freshness and food waste can be found for all food categories, with higher rates of food waste if more loss of freshness was observed. Significant correlations are also found for fruit and vegetables, with more observed loss of freshness for longer storage durations (Table 4).

The ANOVA tests from the linear regression analyses confirm the tendencies of the Spearman correlation with *p* = 0.000 for all tests except for the loss of freshness and storage time for meat and fish with *p* = 0.434, respectively, 0.378. Additional ANOVA tests of between-subjects effects (Appendix A) show significant effects between food waste and the storage time for fruit (*p* = 0.010) and vegetables (*p* = 0.006), implying that both food waste and storage time have an interacting impact on the correlations with the loss of freshness.

Microbial decay was the dominant reason for food waste for fruit and vegetables (25% and 28%, respectively). Additionally, a mushy texture and desiccation contributed to food waste. Off-odour was the main reason for food waste in meat and fish (24% and 23%, respectively) (Figure 4). Additional non-quality-related options and options with minor importance were summed up to the category “others” in the figure.

### 3.4. Consumer Expectations and Demands Related to Refrigerator Performance

Consumer demands in refrigerator storage performance were derived from the evaluation of the most relevant food quality preservation criteria and a rating of the storage features that are perceived as a real benefit in storage support. Extended storage time, food safety and the preservation of taste were the most important performance criteria respondents required for all foods. Storage time showed the highest importance for meat, fish and fruit. Food safety and storage time were rated as equally important for vegetables. Criteria such as the preservation of nutrients scored especially highly for fruit and vegetables. The preservation of food appearance was not rated as relevant (Figure 5).

Prompted options of refrigerator storage features that may be relevant to further improve storage included storage climate features for temperature or humidity control; hygiene features, such as additional hygienic functions and filters for odour or microbes; and features preserving nutrients and taste, such as postharvest illumination or vacuum compartments. From a consumer perspective, overall hygiene functions were rated as most important (36%). This was congruent with the high importance stated by approximately a quarter of respondents for microbial air and odour filtration systems. Storage features with a focus on an optimisation of basic storage climate parameters, such as temperature or humidity control, were rated as important by 27% and 26% of respondents, respectively. Regarding nutrient- and taste-preserving features, more than 40% of consumers claimed that they did not know about the feature (Figure 6).

Significant correlations between food waste and hygienic functions or temperature-controlled compartments can be shown for all food categories and for the filtration of airborne microorganisms for all food categories except fish. In addition, a significant correlation between the humidity-controlled compartment and vegetables is shown (Table 5). Correlations confirm a higher importance of the feature if respondents reported higher food waste rates.

The ANOVA test from the linear regression analyses do not confirm the tendencies of the Spearman correlation except for the importance of hygienic functions and the food waste rate of meat (*p* = 0.007) as well as the importance of air filtration and the food waste rates of vegetables (*p* = 0.002) and fish (*p* = 0.736), respectively.

## 4. Discussion

### 4.1. Consumer Food Storage Behaviour and Refrigerator Feature Requirements

The finding that 51% and 62% of the respondents never waste meat or fish, respectively, and 22% and 25% never waste fruit or vegetables, respectively, is similar to the results from a German study which claims that 58% of the consumers never or almost never waste food [40]. The deviations for fruit and vegetables may be caused by not differing between food categories in that study. Short storage times imply that consumers consume their food shortly after they have bought it. However, the results show different storage behaviour patterns and requirements for fruit and vegetables, on the one hand, and for meat and fish, on the other hand. They also imply advanced requirements from the consumers concerning the preservation of the foods’ quality.

A highly diverse storage behaviour is shown in fruit and vegetable storage: fruit is frequently stored at an ambient temperature and most vegetables within the vegetable drawer, which is in line with other studies [4,19]. Respondents store fruit and vegetables for a longer time period (>80% for about seven days) than all other types of perishable food. The food waste rates for fruit and vegetables are higher than for meat and fish, which has been shown in other studies [13]. Based on the results from this survey, microbial spoilage is the main reason for food waste, followed by the loss of sensory quality parameters, such as desiccation or texture loss. This is in line with a study from the Waste and Resources Action Programme (WRAP) study, which showed that the loss of sensory quality, possibly caused by microbial decay, is the main reason for food waste [20].

Significant correlations are shown for vegetables between the rate of food waste and rate of freshness loss (r_sp_ = −0.299), the rate of food waste and storage time (r_sp_ = −0.103) and the rate of freshness loss and storage time (r_sp_ = 0.094). Significant between-subject effects are found (*p* = 0.006) between food waste and the storage time, which suggests an interdependency between these two parameters, probably due to long storage durations. The results clearly show that a better preservation of vegetables will reduce food waste. Next to temperature, humidity control is the key parameter in the preservation of fruit and mainly vegetables. The main mechanism to reduce desiccation and, thus, loss of texture is to maintain a high relative humidity (rH) of 85–95% rH postharvest [41]. The comparatively low interest of respondents in humidity-controlled compartments, at only 26%, seems to contradict the consumer demand for enhanced storage performance. Contrarily, the weak but significant correlation between the waste rates of vegetables and the importance of these compartments (r_sp_ = 0.069) indicates that consumers with higher rates of vegetable waste would use climate control options. However, improved storage features can only provide a sustainable benefit if an enduring and effective change in consumer attitude is achieved. The lack of knowledge concerning storage and the low priority in food waste reduction behaviours [42] will otherwise counteract the possible storage benefits.

Consumers ask primarily for food safety and other quality criteria that are related to food safety for meat and fish. Microbial decay is shown to be one of the major reasons for food waste (20%) and an additional 40% is related to off-odour, which is a typical indicator of microbial metabolism [43], and a common reason for the disposal of food in general [44]. The results are in line with the increase in food safety concerns in Europe, which has occurred since the mid-1980s, linked mainly to food poisoning [45]. Consumers obviously try to address food safety concerns regarding meat and fish, as shown by the quite distinct storage behaviour patterns of the present survey: Meat and fish are stored for short times, with 60% of meat and almost 80% of fish being consumed within the first three days after purchase, which is in line with other studies [46,47]. As no significant correlations between the storage duration and the loss of freshness are found, but for the storage duration and the rate of food waste (r_sp_ = 0.168–0.169), the conclusion could be that a precaution against suspected microbial decay is a crucial reason for the waste of meat and fish. It is highly likely that the short storage cycles for meat and fish are the root cause for the lack of correlation between freshness loss and food waste, which is also implied by the missing between-subject effects between the storage time and food waste rate.

Performance requirements include all parameters predefined, with storage performance being the most important, followed by microbial safety and taste and vitamin preservation. Since a long storage time also includes the prevention of microbial decay, all results underline the consumer demand for storage conditions that protect perishable food from the growth of pathogens or spoilage germs. The importance of food safety is also clearly shown by the high score of hygiene features (36%). The potential for food waste improvement is shown by the significant correlations between the rate of food waste per type of food and importance of hygiene functions (r_sp_ = 0.134–0.164). The weak but significant correlations between the rate of food waste per type of food, except fish, and importance of the filtration of airborne microorganisms (r_sp_ = 0.084–0.087) imply the same conclusion. The missing significant correlation for fish could be related to the very short storage times for this type of food. Temperature, as the most important storage parameter impacting microbial growth, is of minor importance for consumers (28%), but the potential for an improvement of food waste reduction is indicated by the significant correlation between the rate of food waste per food type and the importance of a temperature-controlled compartment (r_sp_ = 0.097–0.150).

### 4.2. Potentials of Food Storage Improvement within Refrigerators

The results underline that even though the importance of proper handling of highly perishable foods is known to some extent, a high uncertainty and behavioural deficits remain. While Spearman correlations show an increase of the importance of storage features when respondents report higher rates of food waste, the ANOVA does not confirm these results. This implies that the correlations’ results are random, possibly due to the fact that consumers are not really aware of the impact and interactions of the different storage features. In addition, former studies have shown that efforts to ensure correct storage are often neglected [46]. To counteract this behaviour, Ishangulyyev et al. underlined the importance of national and international campaigns to improve the consumers’ awareness of food loss [48]. An example was shown in Italy, where consumers’ awareness of safe storage temperatures increased significantly after frequent lessons aimed to promote healthy behaviour [49]. In addition, a basic interest in inexpensive refrigerator upgrades is present [29]. Since improving domestic refrigerators’ storage performance requires consumer knowledge of proper handling, another approach could be the implementation of nudging measures by manufacturers, such as preset features or colour guides. While consumers show a lack of knowledge regarding storage climate parameters, a basic interest is somehow present, such as in inexpensive upgrades [29]. The data from this survey emphasize the high potential of refrigerator storage compartments that actively support correct storage by preset storage atmosphere conditions along with improved user guidance. Raising the awareness of the key storage climate parameters of temperature and humidity and the benefits of storage compartments providing a proper control of these parameters is a prerequisite to further advance food safety and allow for prolonged storage. Thomas shows a significantly improved storage behaviour by applying additional pictograms for correct storage place identification within the refrigerator [4]. Even though pictograms are currently frequently applied on storage compartments within refrigerators, their meaning may often be unknown or their existence neglected by consumers [20]. Further options may be derived from smart refrigerator appliances in the future: Features such as cameras displaying the refrigerator interior, additionally equipped with image recognition, may further advance storage management or be applied to activate food preservation functions when required.

## 5. Conclusions

The results show that consumers have a high awareness of food safety as they store highly perishable food only for short durations. They also imply the demand for the reduction of food waste, as consumers’ call for longer storage times for food categories with very short storage periods. However, the lack of knowledge regarding the parameters influencing mainly food safety and quality will also have an impact on the correct handling of refrigerator storage options and features. Active guidance would ensure that the features are used properly, thus, supporting the most pressing consumer performance needs of prolonged storage time and food safety. One key element in the future may be smart appliances, providing the consumers with information about the refrigerators contents and related information such as shelf life data.

## Figures and Tables

**Figure 1 foods-09-01462-f001:**
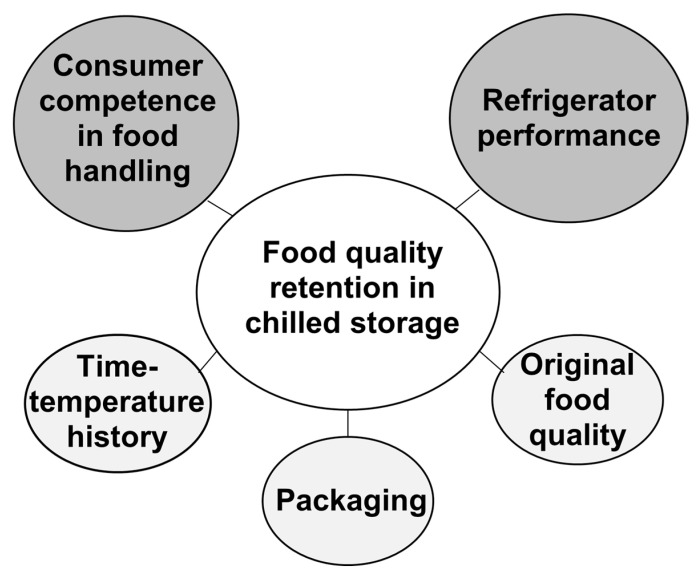
Parameters and their impact on defining food quality retention in chilled storage (own figure; study-focus topics are highlighted in grey).

**Figure 2 foods-09-01462-f002:**
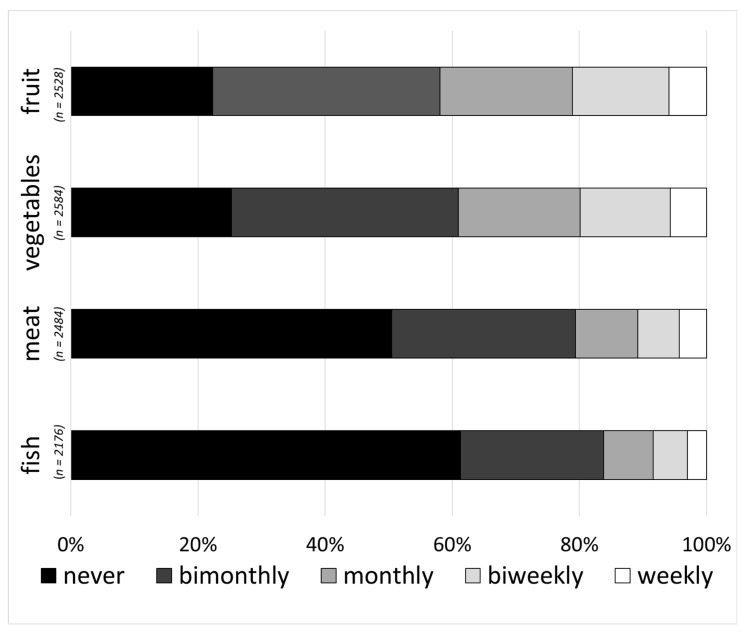
Frequency of respondents’ food waste for fruit, vegetables, meat and fish (multiple-choice, single-answer).

**Figure 3 foods-09-01462-f003:**
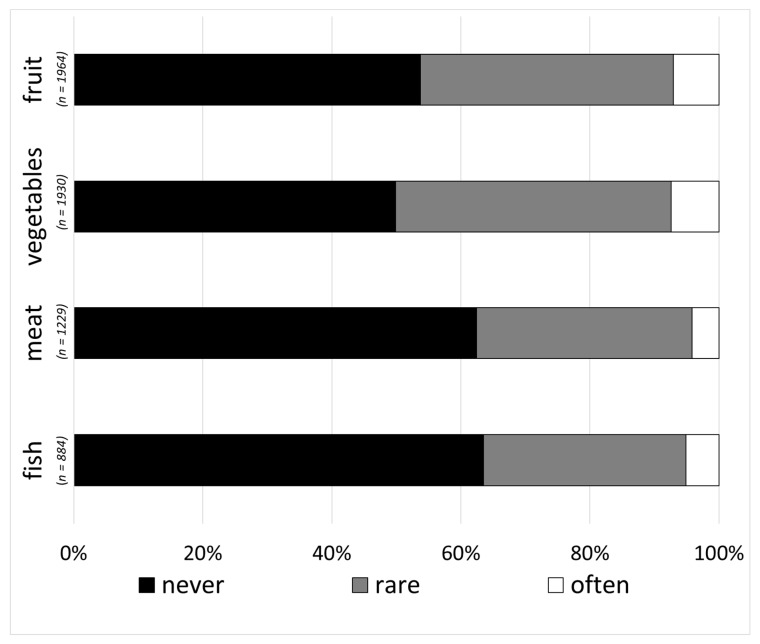
Frequency of respondents’ loss of freshness of fruit, vegetables, meat and fish (multiple-choice, single-answer).

**Figure 4 foods-09-01462-f004:**
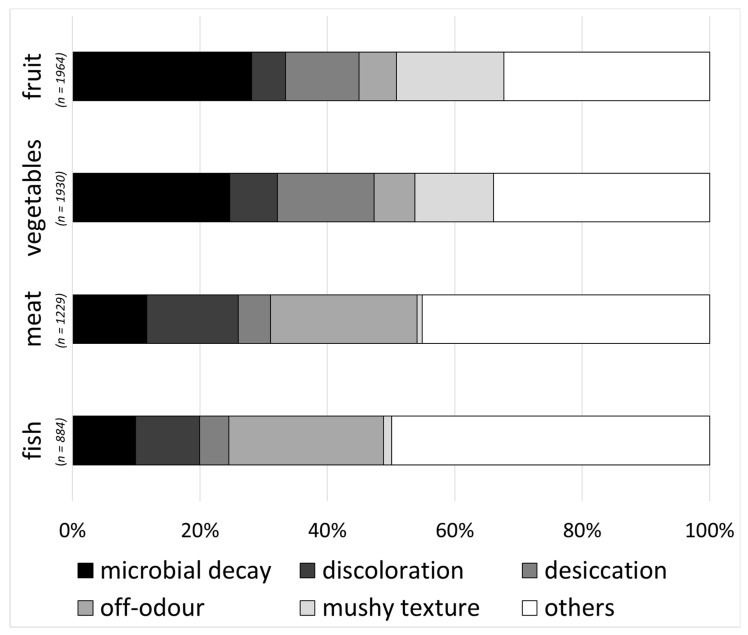
Most common reasons for food waste in fruit, vegetables, meat and fish (multiple-choice, multiple answers).

**Figure 5 foods-09-01462-f005:**
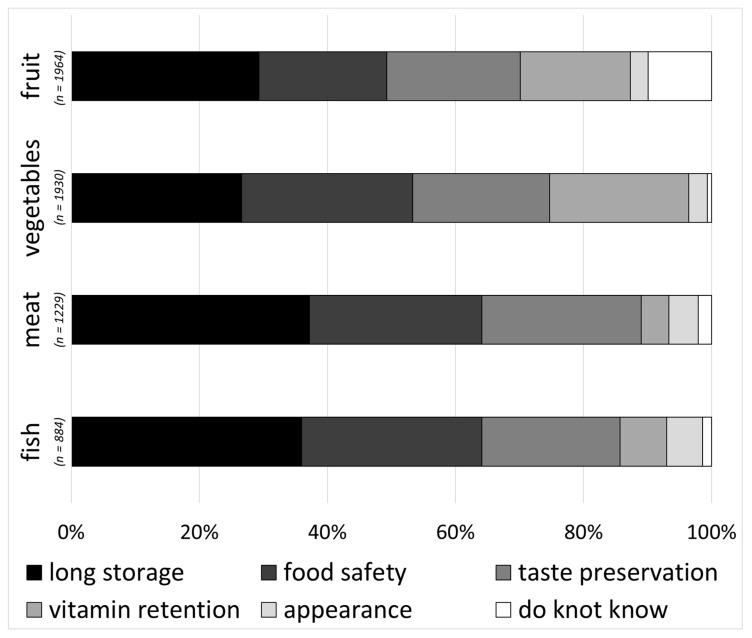
Consumer storage performance requirements for fruit, vegetables, meat and fish (multiple-choice, multiple-answers).

**Figure 6 foods-09-01462-f006:**
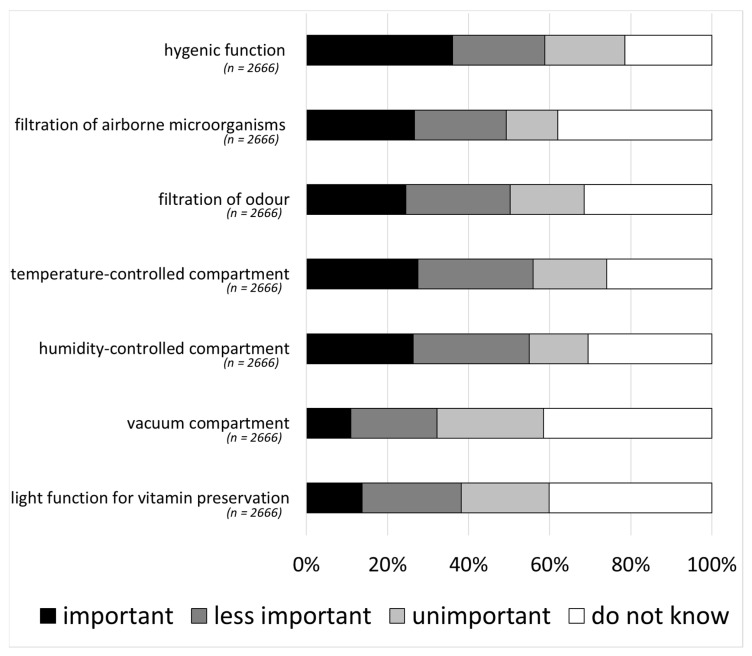
Importance of supplementary storage features (multiple-choice, single-answer).

**Table 1 foods-09-01462-t001:** Number and percentages of demographic and refrigerator characteristics from surveys’ respondents (multiple-choice, single-answer). The data of the population distribution in Germany, 2018, are given as a reference [37,38,39].

Demographic Characteristics	Survey No. of Respondents (*n* = 2666) (%)		Germany (%)
Gender			
Female	1319	49	51
Male	1347	51	49
Age of respondent (yrs.) ^1^			
18–29	417	16	~25 ^2^
30–39	678	25	
40–49	500	19	29
50–59	596	22	
60–69	475	18	/ ^3^
Household size			
1	1276	48	42
2	837	31	34
3	281	11	12
4	203	8	9
5+	69	3	3
Responsibility for food management			
Fully	2027	76	
Jointly	532	20	
Someone else	107	4	
Appliance characteristics			
Age of main refrigerator (yrs.)			
<2	487	18	
2–5	932	35	
6–10	607	23	
>10	540	20	
Do not know	100	4	
Number of refrigerators in household			
1	1492	56	
≥2	1174	44	

^1^ = Survey: Percentages from 2666 respondents; Germany: Percentages from German population (83 Mio), ^2^ = 25 % are related to the age group 20–39 yrs. [38], ^3^ = no data for the age group 60–69 available.

**Table 2 foods-09-01462-t002:** Storage times reported in days and storage places of highly perishable foods (multiple-choice, single-answer, only valid answers included).

Food Categories	Maximum Storage Time (%) (*n* = 2.666)					
	days					
	1	3	5	7	>7	
Fruit	3.2	32.2	33.9	23.7	7.0	
Vegetables	2.3	32.6	34.3	22.9	7.9	
Meat	13.7	46.5	21.4	12.9	5.5	
Fish	37.4	41.6	9.4	5.7	6.0	
		Most Common Storage Place (%) (*n* = 2666)				
	Storage Place					
	Vegetable drawer	Other drawer	Bottom	Mid	Top	Outside
Fruit	42.9	3.2	10.9	7.9	3.5	30.3
Vegetables	81.3	1.2	5.4	5.5	1.4	4.5
Meat	1.8	4.7	32.1	42.4	15.2	2.9
Fish	1.8	5.4	32.4	34.9	15.9	8.3

**Table 3 foods-09-01462-t003:** Correlations between demographic characteristics or storage time and the rate of food waste for food categories (** = significance r_sp_ at 0.01).

Characteristic	Food Waste by Food Category (r_sp_) Weekly/Biweekly/Monthly/Bimonthly/Never			
	Fruit	Vegetables	Meat	Fish
Age groups (yrs.) 18–29/30–39/40–49/50–59/60–69	0.263 **	0.289 **	0.256 **	0.232 **
Household size (persons) 1/2/3/4/5+	−0.140 **	−0.135 **	−0.123 **	−0.121 **
Storage time (d) Max. 1/3/5/7/>7	−0.015	−0.103 **	−0.168 **	−0.169 **

**Table 4 foods-09-01462-t004:** Correlations between the loss of food freshness and the rate of food waste, respectively and the storage time for food categories (** = significance r_sp_ at 0.01).

Characteristic	Loss of Freshness by Food Category (r_sp_) Never/Rare/Often			
	Fruit	Vegetables	Meat	Fish
Food waste Weekly/biweekly/monthly/bimonthly/never	−0.267 **	−0.299 **	−0.258 **	−0.246 **
Storage time (d) Max. 1/3/5/7/>7	0.085 **	0.094 **	0.036	−0.005

**Table 5 foods-09-01462-t005:** Correlations between the importance of food category-related storage features and the rate of food waste (** = significance r_sp_ at 0.01).

Importance of Storage Features Important/Less Important/Unimportant	Food Waste from Food Categories (rsp) Weekly/Biweekly/Monthly/Bimonthly/Never			
	Fruit	Vegetables	Meat	Fish
Hygienic function	0.139 **	0.134 **	0.154 **	0.164 **
Filtration of airborne microorganisms	0.087 **	0.084 **	0.086 **	0.040
Temperature-controlled compartment	0.127 **	0.121 **	0.097 **	0.150 **
Humidity-controlled compartment	/ ^1^	0.069 **	/ ^1^	/ ^1^

^1^ = No correlations were performed, humidity-controlled compartment is commonly used for the quality retention of vegetables.

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
