# Peer review of "Tackling Food Waste: Impact of German Consumer Behaviour on Food in Chilled Storage"

_foods, 2020, doi:10.3390/foods9101462_

Round 1

Reviewer 1 Report

The paper deals with an interesting and timely topic. Overall the paper is well written although I find the results not clearly presented. Tables and graphs requires some adjustement for a better understanding of the readers.

Following my previous comment, I think that, more in detail, the following points should be addressed by authors.

  1. Please amend the abstract in one only paragraph and there are way too many short paragraphs in all the paper.
  2. Line 44, double reference, please amend.
  3. Line 45, what is a-1?
  4. Line 91, please amend the reference.
  5. Line 102, please amend the reference following journal guidelines.
  6. Please modify the tables in order to make them more understandable. In addition, it might be of interest to add the distribution of demographic characteristics of Germany in order to compare your sample with the reference.
  7. Could you please provide more information regarding the data analysis? By reading the paper again, I think that something is missing. Some more articulated statistical analysis after the ANOVAs, as for example a regression or a structural equation modeling could be of interest for the reader. As it is right now, it’s just a description of your questionnaire with no inference at all.

Author Response

Dear Reviewer,

thank you for your comments. They helped me to improve the submitted manuscript. Regrading to your comments:

English language and style are fine/minor spell check required

During the revision, an editor proofread the manuscript

Point 1: Please amend the abstract in one only paragraph and there are way too many short paragraphs in all the paper.

Response 1: The paragraphs from the abstract were removed. Many abstracts from the manuscript were fused, so now most of the remaining abstracts deal with one topic

Point 2 - 4: Line 44, double reference, please amend / Line 45, what is a-1? / Line 91, please amend the reference / Line 102, please amend the reference following journal guidelines.

Response 2 - 4: Now, the double reference is linked with the correct reference. a-1 meant “per year”, this is now written-out in the manuscript. References are amended following the journals guidelines

Point 5: Please modify the tables in order to make them more understandable. In addition, it might be of interest to add the distribution of demographic characteristics of Germany in order to compare your sample with the reference.

Response 5: Due to your comment, I’ve noticed several problems with the comprehensibility of table 2, which were revised. Matching references with data for German demographic characteristics, the survey is based on, were included

Point 6: Could you please provide more information regarding the data analysis? By reading the paper again, I think that something is missing. Some more articulated statistical analysis after the ANOVAs, as for example a regression or a structural equation modeling could be of interest for the reader. As it is right now, it’s just a description of your questionnaire with no inference at all.

Response 6: Material and Methods now include detailed information about the data analysis

Thank you for your help.

Reviewer 2 Report

Dear authors,

I think that the manuscript doesn't give useful information to reduce the waste production, since is only a preliminary analysis on the consumers habits. 

Author Response

Dear Reviewer,

thank you for

your comments. They helped me to improve the submitted manuscript. Still, there are questions about two of your points.

Extensive editing of English language and style required

During the revision, an editor proofread the manuscript

Point 1: You pointed out that the research design, the presentation of the results and the conclusions must be improved.

Response 1: Do you refer each of these points to your criticism about the usefulness of the information for food waste? While we can revise the presentation of the results and the conclusions, the research design was defined by the survey and cannot be changed anymore.

Point 2: I think that the manuscript doesn't give useful information to reduce the waste production, since is only a preliminary analysis on the consumers’ habits. 

Response 2: We think that the results from the survey lead to the depicted conclusions, that a reduction of food waste is possible via offering food-adjusted storage places within refrigerators. To underline the results, we performed ANOVA tests during the revision. The results match with most Spearman correlations but also imply the lack of knowledge by consumers about proper storage conditions.

Thank you for your help.

Reviewer 3 Report

While the topic of this study is interesting, the following things should be clarified. In particular, I look forward to further analysis of the current data to enjoy more insightful findings. 

  1. Abstract: The abstract should be one paragraph. Also, please clarify the following sentences: lines 19-20, 24-25, 27-28, and 28-30. When you describe the results of this study, please use the “past” forms of verbs.
  2. There are many short paragraphs that include one sentence in this manuscript. Please try to make paragraphs that may include multiple sentences by adding more descriptions or combining other sentences or paragraph related.
  3. Lines 38-40: It would be clearer if you define “food waste” and its scope.
  4. Line 41: Please specify the different parameters.
  5. Line 45: Please specify the Year related to the data.
  6. Lines 68-71: If you specify the year related to each data (e.g., Ireland, Greece, UK), it would be easier to follow.
  7. Line 150: Please specify the period of data collection, i.e., from when to when.
  8. Lines 176-180: Please provide more detail information about the data analysis. Why did you use the non-parametric analysis? What were the two factors in the two-way ANOVA? Were there any post hoc testing? This part should be further revised more clearly.
  9. I am wondering if there were no respondents who did not complete the entire questionnaire. If any, please specify them.
  10. Lines 233-237 and Table 3: It would be easier to follow if the authors convert the values of “Food waste by food category” the other way around. For example, it can be listed as never/bimonthly/monthly/biweekly/weekly.
  11. Lines 254-257 and Table 4: In the same manner, if the authors convert the values of “Food waste” the other way around (i.e., never/bimonthly/monthly/biweekly/weekly), it would be easier to read.
  12. Lines 262-263: It is difficult to understand this part. How did you analyze? Please specify it more detail. If you can show the tables at least in the response to the reviewers’ comments, it would be helpful.
  13. Lines 299-303 and Table 5: Same here as I mentioned in the Q10 and Q11.
  14. In the result section, the authors mainly demonstrated the frequency trends of the results. Even though the results are informative, it would be more insightful if they explore other types of statistical analyses as well. Even though the authors did great job in collecting data from 2,666 respondents, their current show is not exciting and interesting enough, indeed. They should spend more time in exploring their data. In addition, using regression models would be more helpful to draw a big picture of this study.
  15. Lines 410-412: I don’t think that this sentence makes sense because there is no data to support in this study.
  16. There is no description about the protocol approval by the Institutional Review Board (IRB) or Ethic committee, although this study is related to human subjects.

Author Response

Dear Reviewer,

thank you for your comments. They helped me to improve the submitted manuscript. Regrading to your comments:

English language and style are fine/minor spell check required

During the revision, an editor proofread the manuscript

Point 1: Abstract: The abstract should be one paragraph. Also, please clarify the following sentences: lines 19-20, 24-25, 27-28, and 28-30. When you describe the results of this study, please use the “past” forms of verbs.

Response 1: The paragraphs from the abstract were removed and the mentioned sentences clarified. Sentences in the abstract about the results are now written in the past form

Point 2: There are many short paragraphs that include one sentence in this manuscript. Please try to make paragraphs that may include multiple sentences by adding more descriptions or combining other sentences or paragraph related.

Response 2: Many abstracts from the manuscript were fused, so now most of the remaining abstracts deal with one topic

Point 3: Lines 38-40: It would be clearer if you define “food waste” and its scope

Response 3: The term “food waste” is now defined as “consumer-related”

Point 4 - 5: Line 41: Please specify the different parameters / Line 45: Please specify the Year related to the data.

Response 4 - 5: The mentioned parameters and years were included and specified

Point 6: Lines 68-71: If you specify the year related to each data (e.g., Ireland, Greece, UK), it would be easier to follow.

Response 6: The years, related to the data were included

Point 7 - 8: Line 150: Please specify the period of data collection, i.e., from when to when / Lines 176-180: Please provide more detail information about the data analysis. Why did you use the non-parametric analysis? What were the two factors in the two-way ANOVA? Were there any post hoc testing? This part should be further revised more clearly.

Response 7 - 8: We included specific information about the period of data collection. Also, Material and Methods now contains detailed information about the data analysis

Point 9: I am wondering if there were no respondents who did not complete the entire questionnaire. If any, please specify them.

Response 9: An institute for consumer research conducted the survey. While some questions could be excluded due to respondents habits (like questions about the storage of meat for vegetarians), we do not know if respondents did not complete the complete questionnaire and cannot provide any numbers

Point 10 – 11; 13: Table 3/4/5: It would be easier to follow if the authors convert the values of “Food waste by food category” the other way around.

Response 10 – 11; 13: While some questions could be excluded due to respondents habits (like questions about the storage of meat for vegetarians), we gain no information by the institute if respondents did not complete the complete questionnaire. Therefore, we cannot provide any numbers

Point 10- 11; 13: Table 3/4/5: It would be easier to follow if the authors convert the values of “Food waste by food category” the other way around.

Response 10 – 11; 13: Is this crucial for you? The institute for consumer research determined the order of the parameter values. Since we have 2666 respondents, converting the order of the values would take time. In addition, we descripted the results from each Spearman correlations in the manuscript.

Point 12: Lines 262-263: It is difficult to understand this part. How did you analyze? Please specify it more detail. If you can show the tables at least in the response to the reviewers’ comments, it would be helpful.

Response 12: The demanded information is now part of Material and Methods

Point 14: In the result section, the authors mainly demonstrated the frequency trends of the results. Even though the results are informative, it would be more insightful if they explore other types of statistical analyses as well. Even though the authors did great job in collecting data from 2,666 respondents, their current show is not exciting and interesting enough, indeed. They should spend more time in exploring their data. In addition, using regression models would be more helpful to draw a big picture of this study.

Response 14: We added linear regressions models, including ANOVA tests. The results are shown in the manuscript and discussed

Point 15: Lines 410-412: I don’t think that this sentence makes sense because there is no data to support in this study.

Response 15: You are right, the sentence was removed

Point 16: There is no description about the protocol approval by the Institutional Review Board (IRB) or Ethic committee, although this study is related to human subjects.

Response 16: The University Bonn, which commissioned the survey, has no ethic committee. While the data are related to humans, they are anonymised and cannot be traced back to the respondents by the authors of the manuscript and therefore not by any readers. We added this information into the manuscript

Thank you for your help.

Round 2

Reviewer 1 Report

The paper has improved from the last version and I thank the authors for addressing my comments. I still see no column related to the German population in table 1 as I asked for, though. 

Author Response

Dear Reviewer,

thank you for your comment from round 2. It helped me to improve the submitted manuscript. Regrading to your comment:

Point 1: The paper has improved from the last version and I thank the authors for addressing my comments. I still see no column related to the German population in table 1 as I asked for, though. 

Response 1: A column for the comparison of German demographic data was included into Tab. 1 (line 193). Tab 1 is part of the chapter “results”, which now includes data that are not based on our research, but on references [37-39] from the Federal Statistical Office of Germany (StBA). The results for age groups between this survey and StBA can only be compared to a limited extent: For the survey, 100 % of the respondents include only the age groups between 18 – 69, while for data from the StBA, 100 % include all ages in Germany.

Thank you for your help.

Reviewer 2 Report

The manuscript was improved and the critical points were clarified.

Author Response

Dear Reviewer,

thank you for accepting the revised version of the manuscript “foods-938580”. The newly revised and uploaded version from round 2 only contains changes that other reviewers required.

Thank you for your help.

Reviewer 3 Report

  1. I hope that I am reviewing a revised version of the original manuscript. Although you revised the abstract, e.g., “Sentences in the abstract about the results are now written in the past form”, the part is still in the present form.
  2. Lines 15-17: Please rephrase this sentence. The word, “therefore,” is an adverb. Thus, it can be changed to either “Perishable foods … in refrigerators. Therefore, both consumer …” or “Since perishable foods … in refrigerators, both consumer …”.
  3. Tables 3, 4, and 5: It does not seem that you captured my point correctly. Why do you think that it takes time to covert? It takes only five minutes to change and revise it. If you don’t know how to do, that is fine. I wanted to help you guide in a reader-friendly manner.
  4. IRB: You mentioned “The University Bonn, which commissioned the survey, has no ethic committee.” However, it seems that your university has the IRB committee: https://www.uni-bonn.de/forschung/Drittmittel%20und%20Projekte/eu-forschungsprojekte/eneri Is it a different university?

Author Response

Dear Reviewer,

thank you for your comments from round 2. They helped me to improve the submitted manuscript. Regrading to your comments:

Point 1: I hope that I am reviewing a revised version of the original manuscript. Although you revised the abstract, e.g., “Sentences in the abstract about the results are now written in the past form”, the part is still in the present form.

Response 1: You received the revised version, but due to some error, the sentences were still in present form. Now, all sentences regarding the results and conclusion are written in the past form

Point 2: Lines 15-17: Please rephrase this sentence. The word, “therefore,” is an adverb. Thus, it can be changed to either “Perishable foods … in refrigerators. Therefore, both consumer …” or “Since perishable foods … in refrigerators, both consumer …”

Response 2: The sentence from Line 15-17 has been changed, following your comment

Point 4: You mentioned “The University Bonn, which commissioned the survey, has no ethic committee.” However, it seems that your university has the IRB committee: https://www.uni-bonn.de/forschung/Drittmittel%20und%20Projekte/eu-forschungsprojekte/eneri Is it a different university?

Response 4: The link you provided is for ENERI “European Network of Research Ethics and Research Integrity”. ENERI ended at the 31th August 2019.

At the University Bonn, only the Faculty of Medicine has an ethic committee (https://ethik.meb.uni-bonn.de/). The survey from this submission was conducted by the Institute of Agricultural Engineering, which is part of the Faculty of Agriculture (https://www.lf.uni-bonn.de/en/institutes?set_language=en). Also, the ethical committee from the Faculty of Medicine does not consider applications, based on quantitative data only, as long all data protection parameters were taken into account.

The University of Applied Sciences Albstadt-Sigmaringen (Baden-Wuerttemberg) has also no ethic committee. The state of Baden-Wuerttemberg has a medical association with an ethic committee (https://www.aerztekammer-bw.de/10aerzte/48ethikkommission/index.html) which is only responsible for clinical studies.

Thank you for your help.
